# Can network attention effectively stimulate corporate ESG practices?—Evidence from China

**En Xie**, **Shuang Cao**\*

International Business School, Hainan University, Haikou, Hainan, China

\* youyou_cs@126.com

## Abstract

Environmental, social, and governance (ESG) has emerged as a widespread concern for all societal segments. This study aims to explore the influence of network attention on corporate ESG practices from an investor perspective. We find that rising network attention significantly increases corporate ESG practices. Specifically, network attention plays the role of external monitoring, image promotion incentives, and mitigation of financing constraints to make companies willing to challenge ESG practices. Additionally, the promoting effect of network attention on firms' ESG practices was more significant in higher marketization processes, severely competitive industries, and non-state enterprises. In the internet era, companies must pay attention to the flow effect caused by network attention, meet stakeholder demands, and pursue long-term sustainable development.

## 1. Introduction

Currently, many new situations including COVID-19 epidemic, four circuit breakers in the US stock market within two weeks, global warming, natural resource depletion, and corporate scandals have raised social expectations towards corporate environmental, social, as well as moral responsibilities [1]. As a major manufacturing country, there is a particularly prominent contradiction between economic growth and sustainable development in China [2]. The 20th National Congress of the Communist Party of China has clearly stated that high-quality development is the primary task of building a Socialist state comprehensively. The concept of environmental, social, and governance (ESG) is highly consistent with the national development strategy and has become an important measure for sustainable development of enterprises, attracting widespread attention from consumers, investors, and other stakeholders. In terms of China's investment structure, there are fewer institutional investors, mainly a large number of retail investors [3]. The attention from individual investors can create certain external pressure for companies, and the participation of retail investors has become an essential part of establishing and improving the ESG system.

With the inevitably rapid development of China's internet technology, the size of China's internet users reached 1.032 billion as of December 2021, with an internet penetration rate of 73%. Retail investors, who are at an information disadvantage, actively search the internet to

**Data Availability Statement:** All relevant data are within the paper and its Supporting information files.

**Funding:** The author received no specific funding for this work.

**Competing interests:** The authors have declared that no competing interests exist.

find out information about companies in order to make accurate investment decisions. The popularization of the internet has enriched the channel for investors to obtain information [4]. Therefore, the search data can reflect the investors' concerns about capital market transactions and demand, which can function as the most direct and objective evidence of retail investors' online attention. Currently, the internet has entered an era where the volume of followers is king, and online attention will gradually become a resource affecting investment decisions of enterprises [5].

Existing research on ESG at home and abroad mainly focuses on the relationship between ESG and financial performance [6, 7], as well as enterprise value [8, 9]. The ESG rating range is wide, and there are also many potential drivers. This paper explored the influencing factors that drive companies to voluntarily choose ESG strategies to answer the question of how companies can achieve sustainable development. In fact, the informal system has long been a neglected and critical variable in driving ESG management and practice [10]. In China's imperfect legal market, network attention, as a kind of informal regulation [4], represents a powerful complement to the formal system and serves an essential role in enhancing the level of corporate governance and protecting stakeholders. In addition, scholars have examined the impact of investor attention on green innovation in enterprises [11]. However, there are few literature studies on the impact of network attention on enterprise ESG practices based on the effect of internet traffic.

In view of this, we empirically examine the impact of network attention on corporate ESG practices. This study discovered that the network attention had positively contributed to corporate ESG practices regardless of whether ESG has been viewed as a whole or broken down into its three sub-dimensions of E, S and G. Simultaneously, online attention possesses external governance effects, predominantly by enhancing corporate ESG practices through the mechanisms of supervisory advantages, image promotion incentives, and alleviation of financing constraints. In further heterogeneity analysis, whether there are differences in the relationship between network attention and ESG at the national, industry, and corporate levels are discussed in this paper. According to the results, in high marketization, strong industry competition, and non-state-owned enterprises, there is more significant positive correlation between network attention and ESG practice.

Our predominant potential contributions are in the following three aspects. Firstly, there is limited research on the impact of online attention on company decision-making behavior. This paper expands the research on the consequences of investors' online attention to economic impact from the financial market to corporate behavior governance, thus enriching the scope and effectiveness of online attention. The Chinese A-share market is the largest developing market in the world, with a large sample of companies verifying the impact of retail investor network attention on corporate ESG behavior. Simultaneously, this study also decomposed ESG into three subcategories (environmental, social, and governance) for validation, which greatly supplements existing literature on a single dimension. Moreover, this paper provides a new perspective for the research on the governance function of network attention, proving that network media play an external governance role like traditional media.

Secondly, we approach from the perspective of informal external institutions, providing a new perspective for identifying the motivations behind ESG behavior in enterprises, which greatly expands the literature volume on factors determining ESG practice in enterprises. It has been confirmed by previous literature that social media [12] and social trust [13] are all effective informal external governance mechanisms when enterprises implement ESG practices. Compared to reports from professional media organizations, investors accept information in a passive way. Online attention is more manifested as investors actively searching for

information. In this paper, the impact of online attention on corporate social environmental governance behavior has been significantly enriched.

Thirdly, in this paper, online attention is perceived as a traffic resource, in which retail investors are actively performing a corporate governance role by creating external regulatory pressure through the internet. Potential new channels for investors to influence company behaviour include two dimensions of willingness and ability. Through external regulatory governance, image promotion incentives, and mitigation of financing constraints, online attention primarily promotes corporate ESG initiatives. Additionally, in the context of China's fundamental national environment, this paper examines the boundary conditions under which network concerns influence corporate ESG performance from the perspectives of the marketization process, product market competition heterogeneity, as well as property rights heterogeneity.

## 2. Literature and hypothesis development

### 2.1 The economic consequences of network attention

Internet search has become the most commonly adopted channel for people to access massive information resources in their daily lives. The online search based investor attention in the capital market is also becoming a research hotspot in the field of behavioral finance. Goodell et al. (2022) [14] summarized the thematic content of investor attention research from 1994 to 2021 through bibliometric analysis. Domestic and foreign scholars' research on the economic consequences of online attention has focused on futures markets, commodity markets, currency markets, etc. [15–18]. For instance, the scholars assumed that the attention of retail investors is positively correlated with the Systematic risk of enterprises [19], and negatively correlated with the risk of stock price crash [20].

In addition, some literature studies have shown that online attention plays an important role in the behavior and decision-making of enterprises. Ang et al. (2021) [21] found that individual investors play a role in corporate governance, and discussions on stock forums can affect merger and acquisition decisions and information disclosure behavior. Cao et al. (2023) [22] assessed that there is a significant U-shaped relationship between investor attention based on online search and earnings management in the capital market. Hao (2023) [23] considered that investor attention can significantly promote corporate innovation. Li et al. (2022a) [4] have empirically demonstrated that the internet can significantly enhance green innovation in enterprises. Deng et al. (2020) [5] proposed that increasing online attention can significantly promote corporate growth. The investor network attention generated by internet traffic has become a new resource, and whether it can promote corporate social environmental governance behavior is still worth exploring.

### 2.2 Factors influencing ESG

There is abundant literature on the single dimensional influencing factors of ESG, and some scholars have carried out literature review based research [24–26]. In addition, researches on ESG synthesis have gradually increased in recent years, which are mainly conducted from multiple perspectives such as macro environment, enterprise characteristics, as well as corporate governance. (1) Factors affecting the macro environment of enterprises: Existing literature suggests that Confucian culture [27], social trust [13], government procurement [28], natural disasters [29], etc. can all promote ESG activities for enterprises. However, the intensification of product market competition has a negative impact on ESG in emerging markets [30]. (2) Factors influencing enterprise characteristics: From the perspective of internal characteristics of the enterprise, company size [31], property rights reform [32], controlling shareholder

pledging [33], as well as other factors can all improve the ESG score of the enterprise. From the perspective of enterprise strategic positioning, enterprises with strong management orientation will have higher ESG performance [34]. Merger and acquisition transactions are of great significance for ESG performance with a one-year lag [35]. (3) Factors influencing corporate governance: Aabo and Giorici (2022) [36] proposed that there is a strong positive correlation between female CEOs and ESG scores. Menicucci and Paolucci (2022) [37] explored the impact of board diversification on ESG performance, according to the empirical result, board size, board independence, as well as the presence of sustainability committees, had a positive impact on ESG performance of banks. Wang, Zhang and Qi (2023) [38] found that the hometown identity of the CEO is associated with higher ESG performance. Wang, Qi and Zhuang (2023) [39] explored the collusive effect of multiple major shareholders (MLS) on corporate ESG performance and it turned out that companies with MLS often have lower ESG performance than companies with a single major shareholder. Many scholars have also studied the impact of external governance mechanisms on ESG performance. For example, Y. Wang et al. (2023) [40] found a significant positive correlation between institutional shareholding and corporate ESG performance. Adhikari (2016) [41] contends that companies with a wide coverage of analysts often have a lower level of social responsibility. (4) Other influencing factors: For example, audit reports [42], stakeholders of the contract [43], etc.

## 2.3 Hypothesis development

As the largest group of investors in China's capital market, retail investors are exposed to the threat of profit encroachment from management and major shareholders. Therefore, they would actively gather more information to make accurate investment decisions. The frequency of their keyword searches will reflect the level of interest in the issue [44]. This paper contends that online attention is able to motivate corporate ESG practices by leveraging the advantages of distinctive monitoring, reputational publicity incentives, and financing constraints to generating an external governance effect. Specifically, it will lead to a better ESG performance through the following three main channels.

Network attention offers the advantage of monitoring curbing myopic behaviors of management, which results in greater corporate ESG willingness. As soon as a company is brought to the attention of more investors and exposed to the public, current trends in corporate development, investment decisions, and personal behavior of executives will all be under the scrutiny of the internet [45]. Investors are not only the recipients of information but also disseminate their attitudes and viewpoints promptly, swiftly creating substantial online public opinion pressure [46]. Subsequently, this pressure will force companies to adapt their behavior patterns to fit social ethics [47]. The principal reason is that ethical business practices enables the company to fortify its legitimacy, consequently enhancing the prospects of sustainable survival [48]. Additionally, online attention can heighten the heat of discussion on key events for listed companies, which may subsequently garner the attention of other external monitoring organizations such as the media, intermediary organizations, and regulators. The intervention of administrative supervision increases the probability of discovering corporate violations which acted as a prior "deterrent" to listed companies. Simultaneously, administrative penalties will heighten the cost of self-interested or opportunistic behavior. Accordingly, managers will cautiously assess the consequences of their non-conforming behavior. Therefore, in order to attain legitimacy, enterprises intensify their engagement in ESG practices.

Secondly, network attention has gradually become an influential traffic resource, which can effectively enhance the exposure rate of listed companies and serve as a publicity role. Internet attention is equivalent to free advertising for a company. The company would actively portray

more positive images when the network attention is high. Management attaches significant importance to the reputational capital and value of the company in the labor market. On the one hand, reputation is an immensely vital intangible asset of a company that can improve its competitiveness. On the other hand, the market may heavily rely on this capital to infer manager capabilities and determine executive compensation [49]. As such, the loss of a company's reputation will inevitably translate into a lower share price, potentially damaging the company's market value [50]. In a highly competitive industry, the corporate management level usually actively engages in ESG activities and views it as an impression management tool. Simultaneously, certain companies even demonstrate pro-social behavior by falsifying and exaggerating propaganda, actively participating in the battle for internet traffic by establishing a positive image to cater to the rise in internet attention. When companies with a high level of online attention encounter negative news, managers commonly tend to engage more in ESG activities to conceal negative company information or minimize damage. Therefore, network attention will incentivize companies to engage in ESG practices, attain reputation capital, and achieve optimal publicity.

Thirdly, network attention can ease corporate financing constraints [51], as well as increase the companies' ability to enhance ESG investment. Investors' profit-seeking mindset and disadvantageous information position would drive them to conduct web searches thus mitigating the effect of information asymmetry between investors and firms. Investors' attention is limited, and their external attention is scarce. Therefore, the more they follow a company, the more likely they are to purchase its shares [52]. This potential investment intent enhances the company's ability to raise equity and continue as a going concern. The high internet "traffic" of online attention will unavoidably generate a wide range of influence and attracts additional stakeholders to the business. With the improvement of corporate social networks, it will be able to assist companies in acquiring more key resources such as human and financial resources. Also, network attention has increased, various types of corporate information have been mined and disclosed, and the transparency of corporate information has significantly increased. In order to deliver high-quality external disclosure content, companies will endeavour to demonstrate, through ESG practices, that they are socially responsible and may be exposed to relatively less environmental, governance and other risks [53]. A company's ESG performance will be able to vigorously represent its past performance over time and is more representative of its market outlook, thereby providing investors' confidence. The optimistic capital market environment has rendered it more accessible for the company to secure finance, thereby providing the necessary financial foundation for ESG practices.

Therefore, the following hypothesis is proposed. H1: network attention improved corporate ESG practices.

## 3. Methodology

Considering ESG performance scores' availability, this study has selected A-share listed companies in China's Shanghai and Shenzhen markets between 2011 to 2021 as the research subjects. The following principles were followed in the processing of data: (1) The financial sector samples are excluded; (2) The ST or PT categories are excluded; (3) The samples that possess missing and abnormal data are excluded. Subsequently, a two-sided Winsorize shrinkage at the 1% level for all continuous variables has been successfully performed to effectively prevent the elimination of extreme observations from affecting the empirical results. The final consists of 9,588 firm-year observations. Simultaneously, the network attention data has been obtained from the CNRDS database, which provides internet search index values for China's listed

companies since 2011. We also successfully acquired the relevant financial data of the company from the China Stock Market and Accounting Research (CSMAR).

### 3.1 Dependent variables

In consideration of the length of the sample as well as the completeness of the indicator system, the composite ESG score from the Bloomberg database and the individual sub-scores for the fundamental three dimensions of the environment (E), social responsibility (S), and corporate governance (G) have been employed as indicators of ESG practice. This score is a comprehensive assessment of the three dimensions of the enterprise' s environment, society, as well as governance based on the corporate social responsibility report, annual report, website and other public information available to investors, and the overall rating of the company is based on the weighted average of the individual scores obtained from the ESG comprehensive score. Correspondingly, the scores will range from 0 to 100, whereby the higher the score, the higher the level of ESG compliance.

### 3.2 Independent variables

The internet search index effectively measures network attention (ATT), a comprehensive search index calculated based on various internet search data on the Baidu platform and integrated with relevant information such as news and public opinion. With reference to the previous research literature [20], the total value of the search sum for keywords such as stock code, company profile, and the full name of the company has been divided by the number of keywords followed by the natural logarithm to represent the web attention.

### 3.3 Control variables

Based on the ESG related literature [24, 54], we choose the following control variables: firm size (Size), firm age (Ln_Age), leverage (Lev), profitability (ROA), management shareholding (MS), the shareholding of institutional investors (INST), share concentration (TOP1), board size (Board), CEO duality (Dual) and corporate property (SOE). All variable definitions are reported in Table 1.

### 3.4 Empirical model

We utilize the following model to validate the causal relationship between network attention and corporate ESG practices. All regression results have reported robust heteroskedasticity standard errors.

$$ESG_{i,t} = \alpha_0 + \alpha_1 ATT_{i,t} + \alpha_2 Controls_{i,t} + \sum Industry + \sum Year + \varepsilon_{i,t} \qquad (1)$$

The independent variable $ATT_{i,t}$ denotes the network attention of firm i in year t. ESG, E, S, and G represent the dependent variable. $ESG_{i,t}$ denotes the total ESG score of firm i in year t or the score of one of the three subscales E, S, and G. $Controls_{i,t}$ represents a set of control variables, $\sum Industry$ denotes the control industry dummy and $\sum Year$ denotes the control time dummy. If the network attention (ATT) coefficient $\alpha$ is positive, then hypothesis 1 holds.

## 4. Results

### 4.1 Descriptive statistics

Table 2 depicts the descriptive statistics of all the variables. The mean value of corporate ESG practices is 27.91. Overall, companies' ESG performance has been relatively poor. Among the

**Table 1. Variable definitions.**

| Variablee | Variable Definition |
|---|---|
| ESG | Total Corporate ESG Performance Score. |
| E | Environmental score for ESG performance. |
| S | Social Responsibility Score for ESG Performance. |
| G | Corporate Governance Score for ESG Performance. |
| ATT | Natural logarithm of the (index value /keyword number). |
| Size | Natural logarithm of the (total asset) |
| Ln Age | Natural logarithm of the year of listing. |
| Lev | Total liabilities divided by total assets. |
| ROA | Net profit divided by total assets. |
| M.S. | Management shareholding data divided by total share capital |
| INST | The percentage of shares hold by the institutional investors. |
| Top1 | The total shareholding ratio of the top ten shareholders. |
| Dual | Equals1if CEO is also the chairman of the board, and 0 otherwise. |
| Board | Number of Directors. |
| SOE | Equals to 1 for state-owned enterprises, 0 otherwise. |
| Industry | According to the "Industry Classification Guidelines" by China Securities Regulatory Commission, I divide the listed companies by sector. |
| Year dummy | Set reference variables based on the years 2011–2021 |

three sub-index scores, the mean value of corporate governance is 63.40, the mean value of social responsibility is 12.16, and the mean value of environmental performance is 8.602, indicating that corporate governance in the sample has been favorable and there is significant room for improvement in environmental investment. The mean value of network attention (ATT) is 12.11, with a standard deviation of 0.675, indicating that differences exist between investors and attention to different companies.

**Table 2. Descriptive statistics of variables.**

| Variable | N | Mean | Std | Min | Max |
|---|---|---|---|---|---|
| ESG | 9558 | 27.91 | 8.553 | 10.70 | 60.60 |
| E | 9558 | 8.602 | 11.87 | 0 | 64.78 |
| S | 9558 | 12.16 | 6.771 | 0 | 40.75 |
| G | 9558 | 63.40 | 13.76 | 29.65 | 89.86 |
| ATT | 9558 | 12.11 | 0.675 | 9.950 | 14.96 |
| Size | 9558 | 23.15 | 1.282 | 20.09 | 26.43 |
| Ln Age | 9558 | 2.466 | 0.708 | 0 | 3.367 |
| Lev | 9558 | 0.475 | 0.200 | 0.0500 | 0.905 |
| Roa | 9558 | 0.0470 | 0.0590 | -0.379 | 0.233 |
| MS | 9558 | 0.0720 | 0.150 | 0 | 0.679 |
| INST | 9558 | 0.545 | 0.224 | 0.0120 | 0.932 |
| Top1 | 9558 | 0.377 | 0.159 | 0.0810 | 0.758 |
| Dual | 9558 | 0.208 | 0.406 | 0 | 1 |
| Board | 9558 | 8.996 | 1.786 | 5 | 15 |
| SOE | 9558 | 0.518 | 0.500 | 0 | 1 |

## 4.2 Regression analysis

Table 3 reports the results of the impact of network attention on ESG practices. Column (1) reports the estimated network attention coefficient of 0.850, which is significant at the 1% level. In (2), by utilizing the company's reported environmental score E as the dependent variable, the results indicate that an increase in network attention will lead companies to place more emphasis on environmental performance. This has been consistent with the previous findings [55]. In column (3), the regression results in a coefficient of investor attention of 0.337, which is significant at the 1% level. This indicates that increased network attention can motivate companies to engage in more socially beneficial behaviors and are vital participants in the country's development and society. In (4), for corporate governance as the dependent variable, the estimated coefficient of network attention is positively related to G with a coefficient of 0.783, which is significant at the 1% level. This indicates that network attention will effectively motivate the company management to emphasize stakeholders' demands more and strengthen corporate governance in general to maximize shareholders' interests under external supervision. Therefore, it is inevitable that the increase in network attention will lead company

**Table 3. Principal regression.**

| Variables | ESG | E | S | G |
|---|---|---|---|---|
|  | (1) | (2) | (3) | (4) |
| ATT | 0.850*** | 1.036*** | 0.337*** | 0.783*** |
|  | (0.121) | (0.213) | (0.130) | (0.162) |
| Size | 1.905*** | 2.748*** | 1.548*** | 1.386*** |
|  | (0.077) | (0.134) | (0.081) | (0.114) |
| Ln_Age | 0.304*** | -0.030 | -0.874*** | 0.905*** |
|  | (0.117) | (0.203) | (0.139) | (0.171) |
| Lev | -2.485*** | -3.188*** | -3.250*** | -2.784*** |
|  | (0.418) | (0.718) | (0.461) | (0.611) |
| Roa | -0.858 | -0.465 | -1.626 | -4.251*** |
|  | (1.121) | (2.011) | (1.222) | (1.563) |
| MS | 5.907*** | 8.719*** | 5.148*** | 5.444*** |
|  | (0.639) | (1.094) | (0.811) | (0.980) |
| INST | 6.774*** | 8.676*** | 3.421*** | 8.224*** |
|  | (0.489) | (0.836) | (0.549) | (0.744) |
| Top1 | -3.768*** | -4.986*** | -3.006*** | -3.808*** |
|  | (0.523) | (0.893) | (0.572) | (0.757) |
| Dual | 0.037 | 0.103 | -0.522*** | 0.389* |
|  | (0.152) | (0.265) | (0.169) | (0.226) |
| Board | 0.024 | -0.003 | 0.027 | 0.026 |
|  | (0.038) | (0.065) | (0.042) | (0.056) |
| SOE | 0.252* | -0.118 | 0.980*** | 0.011 |
|  | (0.152) | (0.265) | (0.159) | (0.210) |
| Constant | -37.692*** | -77.113*** | -30.899*** | 3.958 |
|  | (1.811) | (3.102) | (1.915) | (2.589) |
| Year/ Industry F.E. | Yes | Yes | Yes | Yes |
| Observations | 9,558 | 9,558 | 9,558 | 9,558 |
| R-squared | 0.576 | 0.340 | 0.195 | 0.649 |

Notes:

***, **, and * mean significance level at the 1, 5 and 10%, respectively. Values in parentheses are heteroskedasticity robust standard errors.

managers to be more forward-looking in their strategy formulation and more conducive to the long-term sustainability of the company's growth. Hence, key hypothesis 1 has been fully validated.

## 4.3 Robustness test

**4.3.1 Instrumental variable regression.**　When corporations demonstrate strong performance in ESG, investors tend to exhibit heightened online attention. Consequently, in this research, there is may exist endogeneity issues arising from the potential reverse causality between the independent variable and the dependent variable. We refer to [56] for selecting instrumental variables and use the proportion of shares outstanding (Trade) as the instrumental variable of network attention. In terms of correlation, as the proportion of publicly traded shares increases, the magnitude of attention received by the general public also expands. In terms of exogeneity, the proportion of publicly traded shares is primarily influenced by the early stages of equity split reform and later by the provisions of the Securities Law. The decision of a company to engage in ESG practices is independent of the proportion of publicly traded shares. Table 4 column (1) displays the results of the first stage regression, where shares outstanding (Trade) and network attention are positively correlated, further demonstrating the relevance of the instrumental variables. Column (2) indicate that network attention remains significantly and positively associated with ESG practices after controlling for endogeneity issues.

**4.3.2 Replace key variables.**　We use the summed logarithm of search values for keywords such as stock code, company profile, and full company name from the Web search index database as an alternative measure of network attention (IA). Table 4 column (3) presents the estimated coefficient for network attention as 0.955, demonstrating its statistical significance at

**Table 4. Robustness test.**

| Variables | ATT | ESG | ESG | ESG | ESG |
|---|---|---|---|---|---|
| | **(1)** | **(2)** | **(3)** | **(4)** | **(5)** |
| ATT | | 8.928*** | | | 0.435** |
| | | (1.870) | | | (0.137) |
| Trade | 0.207*** | | | | |
| | (0.030) | | | | |
| IA | | | 0.955*** | | |
| | | | (0.125) | | |
| ATT1 | | | | 1.033*** | |
| | | | | (0.137) | |
| Other Controls | Yes | Yes | Yes | Yes | Yes |
| Constant | 6.646*** | -88.924*** | -38.004*** | -39.577*** | -13.933*** |
| | (0.149) | (12.897) | (1.784) | (1.981) | (2.793) |
| Year F.E. | Yes | Yes | Yes | Yes | Yes |
| Industry F.E. | Yes | Yes | Yes | Yes | No |
| Code F.E. | No | No | No | No | Yes |
| Observations | 8,553 | 8,553 | 9,558 | 8,075 | 9,558 |

Notes:

***, **, and * mean significance level at the 1, 5 and 10%, respectively. Values in parentheses are heteroskedasticity robust standard errors. Due to space constraints, the results of the robustness tests for the three ESG sub-dimensions are not reported but are kept for reference.

the 1% level. This finding indicates that online attention stimulates corporate ESG practices, thereby supporting Hypothesis 1. This result demonstrates the robustness of the findings.

**4.3.3 Change lag time.**   Firms need time to adjust their investment policies dynamically, so this study uses lagged one-period network attention and related control variables, which can also mitigate potential endogenous problems. From the results in Table 4 column (4), it can be observed that the coefficient for ATT1 is 1.033, which is significant at the 1% level. This finding indicates a significant positive correlation between online attention and ESG practices, providing support for Hypothesis 1 presented in this study.

**4.3.4 Control individual fixed effects.**   To mitigate the potential problem of omitted variables that may arise from ignoring firm-specific characteristics, Eq (1) is re-estimated using a model that includes firm-fixed and year-fixed effects. Table 4 column (5) reveal that the ATT coefficient is estimated to be 0.435, demonstrating statistical significance at the 5% level. This observation aligns with and reinforces the previous empirical evidence, providing additional robust support for Hypothesis 1.

## 5. Further analysis

### 5.1 Heterogeneity of the marketization process

China's comprehensive national power continues to develop and is progressively transitioning from a planned economy to a market economy, which is a significant institutional background influencing enterprise micro-behavior. On the one hand, higher marketization implies less government intervention and a high level of investor protection. Therefore, retail investors will actively participate in the capital market, seek information via the network, and increase the company network's attention. As such, companies will implement more ESG initiatives to reduce financing costs and demonstrate the value of long-term corporate growth. On the other hand, regions with a higher marketization process typically possess a more developed legal environment and greater information transparency. Hence, investors will have higher expectations for corporate ESG and can effectively monitor and regulate corporate behavior through online attention. We employ the marketization index developed by Wang, Fan and Hu (2019) [57] to determine each location's marketization level effectively. In Eq. (1), we introduce an interaction term (ATT×Market) to investigate the moderating effect of the marketization process comprehensively.

The results in column (1) of Table 5 demonstrate that the coefficient of the cross-product term is 0.245 and is significant at the 1% level. This implies that the higher the marketization process, the larger the positive influence of network attention on ESG practices.

### 5.2 Heterogeneity of industry competition

First, in the low-competition industry, the market concentration is high and some companies are in a monopoly position. Companies face less competitive pressure and have more stable capital and resources. When the network attention is raised, it has more strength to make ESG investments. Second, when an industry is concentrated in a few large enterprises, other businesses have fewer opportunities to gain market share, while monopolies have more opportunities to gain network attention. Companies will take advantage of the opportunity to increase web traffic and investor recognition by engaging in ESG activities for corporate promotion. Finally, in low-competition industries, where the degree of information misalignment is greater, network attention, an external governance mechanism, may play a greater role and better stimulate corporate ESG practices. This paper uses the HHI, the cumulative square of the ratio of each firm's primary business revenue to the industry's total primary business revenue, to assess the level of competition to which a firm is subject. A higher value indicates that

Table 5. The moderating effect of the marketization process.

| Variables | ESG | E | S | G |
|---|---|---|---|---|
| | (1) | (2) | (3) | (4) |
| ATT | 0.819*** | 0.980*** | 0.227* | 0.784*** |
| | (0.120) | (0.205) | (0.129) | (0.181) |
| ATT×Market | 0.245*** | 0.415*** | 0.244*** | 0.170** |
| | (0.053) | (0.090) | (0.056) | (0.079) |
| Market | 0.436*** | 0.489*** | 0.354*** | 0.500*** |
| | (0.037) | (0.061) | (0.043) | (0.056) |
| Other Controls | Yes | Yes | Yes | Yes |
| Constant | -35.973*** | -71.140*** | -27.898*** | 0.986 |
| | (1.835) | (3.073) | (1.956) | (2.813) |
| Year/ Industry F.E. | Yes | Yes | Yes | Yes |
| Observations | 8,848 | 8,848 | 8,848 | 8,848 |
| R-squared | 0.543 | 0.277 | 0.170 | 0.627 |

Notes:

\*\*\*, \*\*, and \* mean significance level at the 1, 5 and 10%, respectively. Values in parentheses are heteroskedasticity robust standard errors.

there is less industry competition. Table 6 demonstrates that ATT×HHI coefficients are all significantly positive, indicating that the effect of network attention on improving ESG practices is more pronounced in industries with little competition.

### 5.3 Heterogeneity of property rights

In the Chinese institutional environment, where the nature of property rights is an essential characteristic of firms, we further examine whether differences existed in network attention about corporate ESG practices among SOEs and non-SOEs. On the one hand, as SOEs possess a strong political connection with the government and bear the responsibility of maintaining social stability as well as enhancing people's well-being, thus they are required to set an exemplary example of ESG practices at the national level. Hence, the ESG performance of SOEs is often political and mandatory, and regardless if they receive more attention from investors, the ESG investment of SOEs is comparatively positive. Non-SOEs, whose main business goal is to improve economic efficiency, are hardly willing to take the initiative to undertake ESG practices. After all, this investment will not bring objective benefits to the enterprise in the short term. However, when network attention is raised, non-state companies are more proactive in engaging in ESG to gain legitimacy and cater to government policy guidance. Hence, we can infer that ESG practices are more pronounced among non-SOEs when network attention is raised. The ESG×SOE cross-product term is added to Eq (1) and then regressed. Table 7 presents that the regression coefficients of the cross-product terms are significantly negative, except for column (4), which is not significant, which is consistent with the inference that property rights possess a negative moderating effect.

## 6. Discussion and conclusions

### 6.1 Concluding remarks

Network attention has played an important informal external governance role in affecting corporate investment behavior. Based on 9588 firm-level observations in China from 2011 to 2021, we attempt to assess the potential impact of network attention on ESG. The following

**Table 6. Regulating the role of industry competition.**

| Variables | ESG | E | S | G |
|---|---|---|---|---|
| | (1) | (2) | (3) | (4) |
| ATT | 0.821*** | 0.991*** | 0.311** | 0.765*** |
| | (0.121) | (0.213) | (0.130) | (0.162) |
| ATT×HHI | 3.432*** | 5.411*** | 3.352*** | 1.769* |
| | (0.664) | (1.218) | (0.698) | (0.915) |
| HHI | -0.485 | -1.943 | -1.660 | 1.955 |
| | (1.007) | (1.715) | (1.048) | (1.709) |
| Other Controls | Yes | Yes | Yes | Yes |
| Constant | -37.240*** | -76.154*** | -30.221*** | 3.742 |
| | (1.820) | (3.120) | (1.928) | (2.602) |
| Year/ Industry F.E. | Yes | Yes | Yes | Yes |
| Observations | 9,554 | 9,554 | 9,554 | 9,554 |
| R-squared | 0.577 | 0.341 | 0.197 | 0.650 |

Notes:

\*\*\*, \*\*, and \* mean significance level at the 1, 5 and 10%, respectively. Values in parentheses are heteroskedasticity robust standard errors.

significant conclusions are drawn: First, network attention improves corporate ESG practice significantly. Furthermore, network attention is significantly and positively related to all three major ESG sub-dimensions, namely, E, S, and G. Second, the network attention on promoting corporate willingness and ability to practice ESG through external monitoring, image enhancement incentives, and alleviation of financing constraints. Finally, in a higher marketization process, severely competitive industries and non-state enterprises, the influence of network attention on corporate ESG practices has been greater.

**Table 7. Regulating the role of property rights.**

| Variables | ESG | E | S | G |
|---|---|---|---|---|
| | (1) | (2) | (3) | (4) |
| ATT | 1.170*** | 1.618*** | 0.537*** | 0.786*** |
| | (0.158) | (0.283) | (0.172) | (0.199) |
| ATT×SOE | -0.639*** | -1.162*** | -0.398* | -0.004 |
| | (0.190) | (0.329) | (0.205) | (0.266) |
| SOE | 7.991*** | 13.960*** | 5.799** | 0.064 |
| | (2.294) | (3.975) | (2.479) | (3.206) |
| Other Controls | Yes | Yes | Yes | Yes |
| Constant | -41.968*** | -84.893*** | -33.561*** | 3.928 |
| | (2.248) | (4.000) | (2.414) | (3.028) |
| Year/ Industry F.E. | Yes | Yes | Yes | Yes |
| Observations | 9,558 | 9,558 | 9,558 | 9,558 |
| R-squared | 0.576 | 0.341 | 0.195 | 0.649 |

Notes:

\*\*\*, \*\*, and \* mean significance level at the 1, 5 and 10%, respectively. Values in parentheses are heteroskedasticity robust standard errors.

## 6.2 Theoretical contributions and managerial implications

A fundamental framework for corporate sustainability has been supplied by the global ESG system, and the incorporation of ESG into company management and investment decisions has emerged to be an integral component of corporate sustainability practices. ESG goes beyond the conventional financial performance indicators by instinctively evaluating a company's environmental, social, and corporate governance initiatives [24]. ESG emphasizes the unified development of environmental, social, and governance. Berg, Kölbel and Rigobon (2020) [58] argue that the terms ESG and CSR are sufficiently comparable in meaning to be used interchangeably. We view ESG as an extension and enrichment of CSR, but with certain variations, with ESG possessing a more explicit and specific connotation. One notable difference is that ESG explicitly incorporates governance, whereas CSR only encompasses indirect governance issues related to environmental and social considerations [24].

In order to enhance the long-term sustainable development of businesses, it is crucial that companies comprehend the driving forces underlying ESG practices. This paper complements the factors that influence business ESG activities from the informal institutional perspective. It demonstrates that media attention and online attention are equally crucial informal external governance mechanisms for corporations, supplementing to the body of research on external governance for ethical corporate investing practices. Despite the fact that the literature has devoted an abundance of emphasis on media and sustainability issues, for instance by claiming that media attention enhances CSR [59]. According to a study, media attention considerably improves business environmental performance, according to [47]. Online attention, in contrast to traditional media, is comprehensive and swift in nature and indicates an active search behavior of investors. As a result, the cost for investors to investigate and conduct searches for business information has dropped substantially due to the rapid expansion of the digital network [4]. Therefore, it renders it more straightforward for individuals to utilize the internet to disseminate information and achieve specific traffic effects.

On the basis of stakeholder theory, this paper emphasizes the influence of investors as key stakeholders in shaping organizational behaviour. The rapid development and penetration of the internet in China have provided a prime opportunity for this paper to examine network concerns. The stock market has been the primary focus of previous research on the economic consequences of network issues, with corporate governance and the influence on ESG garnering a disproportionate amount of attention in the literature. This study, in accordance with a hypothesis (H1), demonstrates that investor attention significantly stimulates business ESG performance and is positively connected with each of the dimensions (E, S, and G). Scholars are studying the connection between web search volume and corporate environmental performance [55], as well as environmental information disclosure [44], but only one of the environmental dimensions of ESG practices is taken into consideration. Also, their findings are supported by our results, which manifest that network attention is significantly positive for E.

Data above has already proven that there is a causal relationship between network attention and enterprise ESG performance. Under such circumstance, how does online attention motivate ESG practices? Firstly, online attention has a supervisory advantage and can effectively improve the corporate governance environment. Managers are willing to actively engage in ESG practices, so as to reduce unethical and short-sighted behavior, thus gaining investor support. Secondly, the traffic effect brought by online attention [5] motivates ESG practitioners to engage in positive image promotion. Finally, the high traffic brought about by online attention can broaden channels, transmit information, and result in tilted credit resources. Moreover, abundant funds can help enhance enterprise ESG capabilities. In this paper, the functional boundaries of network attention can be evaluated from a heterogeneity perspective.

Combining the actual reform situation and institutional characteristics of China, differences in the role of network attention in enterprise ESG practice is analyzed under different scenarios such as external marketization process, product market competition, as well as property rights nature. This has been instrumental in achieving a comprehensive understanding of the external governance role of network concerns and complements the literature on the external governance of the presence of investor concerns on firms [60].

This study has important managerial implications. For enterprises, they cannot merely pursue the goal of "profit maximization" for businesses while ignoring long-term development. Companies should actively engage in ESG practices as a "value investment" that earns them a reputation and their stakeholders' trust and support. At the government level, they should actively increase network platform construction and strengthen network ecology guidance and governance. Finally, enterprise managers should pay attention to the traffic effect of network attention, make good use of internet platforms, maintain the image and reputation of the enterprise, and transform the pressure of network attention into the driving force of enterprise ESG transformation.

## Supporting information

**S1 Data.**
(ZIP)

## Author Contributions

**Writing – original draft:** En Xie.

**Writing – review & editing:** En Xie, Shuang Cao.

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
