## [Decision Letter · Decision Letter 0]

1 Jun 2023

PONE-D-23-10880Can network attention effectively stimulate corporate ESG practices? — Evidence from ChinaPLOS ONE

Dear Dr. xie,

Thank you for submitting your manuscript to PLOS ONE. After careful consideration, we feel that it has merit but does not fully meet PLOS ONE’s publication criteria as it currently stands. Therefore, we invite you to submit a revised version of the manuscript that addresses the points raised during the review process.

We look forward to receiving your revised manuscript.

Kind regards,

Ercan Özen, PhD

Academic Editor

PLOS ONE

Additional Editor Comments:

Dear Author, Please fix all the problems that stated by the reviewers. The paper need to be improved significately

Reviewers' comments:

Reviewer's Responses to Questions

**Comments to the Author**

1. Is the manuscript technically sound, and do the data support the conclusions?

Reviewer #1: Partly

Reviewer #2: Yes

2. Has the statistical analysis been performed appropriately and rigorously? 

Reviewer #1: No

Reviewer #2: Yes

3. Have the authors made all data underlying the findings in their manuscript fully available?

Reviewer #1: Yes

Reviewer #2: Yes

4. Is the manuscript presented in an intelligible fashion and written in standard English?

Reviewer #1: No

Reviewer #2: Yes

5. Review Comments to the Author

Reviewer #1: The study is generally weak and has weaknesses regarding the interpretation of findings, revealing the importance of the subject, revealing detailed findings, discussion of findings, and conclusion. In addition, robustness checks are appropriate, but regression analysis makes the subject examination weak. The study is open to development.

Reviewer #2: Information about the method used should be given in the methodology section.

Information should be given about how control variables are determined and how they are selected. The control variables used can be supported by attribution.

6. PLOS authors have the option to publish the peer review history of their article (what does this mean?). If published, this will include your full peer review and any attached files.

Reviewer #1: No

Reviewer #2: No

---

## [Author Response · Author response to Decision Letter 0]

4 Aug 2023

Dear editor and reviewers:

Thank you for offering us an opportunity to improve the quality of our submitted manuscript (Can network attention effectively stimulate corporate ESG practices? — Evidence from China). I thank the reviewers for their critical assessment of our work. I have tried our best to improve and made some changes in the manuscript. In the following, I will address their concerns point by point. Thanks a lot for your reviewing. I'm sorry that some pictures and models are not easy to edit in TXT, can you see the word version uploaded by the system? The word version respond to reviewers in more detail.

Responds to the reviewers' comments：

Reviewer 1: 

Comment: The study is generally weak and has weaknesses regarding the interpretation of findings, revealing the importance of the subject, revealing detailed findings, discussion of findings, and conclusion. In addition, robustness checks are appropriate, but regression analysis makes the subject examination weak. The study is open to development.

Reply：Thanks very much for the reviewer's feedback. I have endeavoured to amend the various sections of the paper such as the introduction, literature review, theoretical assumptions, data results and discussion of the results in an attempt to make this paper not too feeble in general.

In response to the questions raised by the reviewer, I have made specific improvements in the following aspects. Simultaneously, I will extract every detailed question and attach specific screenshots of each question for the content in the paper, and a more detailed responses will be provided one by one.

1. Weaknesses regarding the interpretation of findings. I have reexamined and provided further elucidation on empirical results in the manuscript.

2. Revealing the importance of the subject. ① I have revised the introduction section and streamlined the relevant content of the introduction, striving to introduce the research topic of this article based on your opinions, so as to highlight the significance of the topic research. For example, the introduction of ESG based knowledge and policy has been removed. “The United Nations Global Compact (UNGC) first introduced the concept of ESG in June 2004, advocating that companies should consider the unified development of the environment, society and governance while focusing on business, going beyond the classic financial performance indicators. In March 2021, China proposed the “double carbon” target of achieving "peak carbon" by 2030 and "carbon neutrality" by 2060. Simultaneously, the executive meeting of the State Council also proposed to strengthen the quality of listed companies further, perfect the governance rules of listed companies and enhance the transparency of information as well as the quality of disclosure.”

② I added an introduction to ESG related research in the third paragraph of the introduction, summarizing the current status of ESG research. In addition, I emphasized that online attention, as an informal regulation, is a powerful supplement to formal institutions and has a significant impact on ESG practices in enterprises, therefore, this paper is of certain research value from a theoretical background perspective. (Line 48-60)

③ To make the story clearer, the adjusted introduction part is divided into seven paragraphs. The first paragraph introduces the practical background and proposes that the participation of retail investors is an essential part of establishing and improving the ESG system. In the second paragraph, the issue that network attention has a significant impact on the ESG strategic choice of enterprises is raised in this paper. I hope to make a good transition in the practical background introduction, linking the network focus of this paper with enterprise ESG practice, so that the topic of this paper is of certain significance. In the third paragraph, I hope to further elaborate on the importance of this paper in theory as much as possible. In the imperfect legal market in China, there is relatively little research on the impact of online attention as an informal regulation on corporate ESG practices. In brief, the research question raised is whether network attention with internet traffic effects will affect enterprise ESG practices, which fills the gap. (Line 21-60)

3. Revealing detailed findings . ① To reveal detailed research results, I have updated the data in the paper to 2021 based on the reviewer's suggestion. Due to ESG and internet attention to database disclosure data year restrictions. The final sample period of this paper was selected from 2011 to 2021, and the sample size has increased from 7633 to 9558 now. There are slight changes in terms of the coefficients of all regression results in the article, however, the main conclusions remain unchanged and still support the original hypothesis. For example, the main regression coefficients were previously shown in Figure 1 and Figure 2 was obtained after adjustment.

② The introduction about the research conclusions of this article was not very clear, therefore, it has been rewritten. Due to China's special system and the different external environments faced by enterprises, it is worth further discussing whether there are different impacts of online attention on enterprise ESG in different scenarios. I provided a detailed theoretical introduction and empirical test of the heterogeneity research findings in the fifth part, therefore, a summary of the results was presented in the introduction. (Line 67-72)

③ In terms of possible innovations in this paper, I have modified it to discuss the main arguments and findings of this paper. (Line 95-104)

④ In the discussion section, I have supplemented the discussion of the findings in this paper. I am unsure if the discussion is sufficient, and I look forward to your review and suggestions. (Line 484-500)

4. Revealing detailed findings, discussion of findings. I have rewritten the findings and discussion sections. (Line 441-500)

5. Robustness checks are appropriate, but regression analysis makes the subject examination weak. In addressing the crucial aspect of robustness, the present study acknowledges the inadequacies in the treatment of this matter in the original manuscript. In an earnest attempt to enhance its quality, the author has meticulously redeveloped the section pertaining to the explanation of robustness testing outcomes. (Line 318-359)

Reviewer 1

Comment 1: There is a short form of the ESG phrase in the Abstract but there is no long form of the ESG phrase though it is mentioned the first time in the Abstract.

Reply 1: I’d really appreciate the reviewer’s suggestions. Due to my negligence of these details, ESG is not in its long form in when it appears at the first time. I have corrected it in the abstract, details are as follows: “Abstract: Environmental, social, and governance (ESG) has emerged as a widespread concern for all societal segments. ” (Line 9)

Comment 2: The introduction section should be developed in light of the literature. 

Reply 2: The reviewer’s opinion is very pertinent. I spared no efforts to improve the storyline and carefully revised the introduction in light of the literature, so as to highlight the importance of the theme. 

To make the story clearer, the adjusted introduction part is divided into seven paragraphs. In the first paragraph, I argued that the participation of individual retail investors is an indispensable element in establishing a robust ESG framework. I aim to establish a meaningful connection between the online attention and the corporate ESG transformation, thus imparting substantive value and significance to the chosen topic from a real-world perspective. In the second paragraph, I further emphasize online attention will gradually become a resource affecting investment decisions of enterprises . In the third paragraph, I hope to further elaborate on the importance of this paper in theory as much as possible. In the imperfect legal market in China, there is relatively little research on the impact of online attention as an informal regulation on corporate ESG practices. In brief, the research question raised is whether network attention with internet traffic effects will affect enterprise ESG practices, which fills the gap. (Line 21-104 )

Comment 3: The long form of the ESG phrase should be added since it is first mentioned in the manuscript. 

Reply 3: Based on the reviewer’s comments, I have remembered that the long form of ESG phrases should be added when ESG is first mentioned in the abstract and main text. I have made modifications in the first paragraph of the main text, details are as follows: “The concept of environmental, social, and governance (ESG) is highly consistent with the national development strategy and has become an important measure for sustainable development of enterprises, attracting widespread attention from consumers, investors, and other stakeholders.” (Line29-32 )

Comment 4: This is not formal language or academic writing for the literature. Believe phrase is not suitable.

Reply 4: ① According to the reviewer’s opinion, there is a lack of clear and accurate expression in academic writing in the paper. Firstly, in response to the inappropriate use of the word ‘believe’, I have made three changes in this paper. For example: “Simultaneously, online attention possesses external governance effects, predominantly by enhancing corporate ESG practices through the mechanisms of supervisory advantages, image promotion incentives, and alleviation of financing constraints. ” (Line 64-67 )

② Simultaneously, I carefully polished the paper and worked hard to achieve accurate expression. If you feel that there are still problems when reading it, I will further refine and revise to improve my English writing skills.

Comment 5: Who belongs to these results? Is there any proof for this interpretation in the literature?

Reply 5: As far as I am concerned, there may be ambiguity in my expression in this part because I have not expressed it accurately. I mainly want to express that this study evaluates the functional boundaries of network attention from the perspective of heterogeneity. As far as I am concerned, there may be ambiguity in my expression in this part because I have not expressed it accurately. I mainly want to express that this study evaluates the functional boundaries of network attention from the perspective of heterogeneity. As far as I am concerned, there may be ambiguity in my expression in this part because I have not expressed it accurately. I mainly want to express that this study evaluates the functional boundaries of network attention from the perspective of heterogeneity. I conducted theoretical discussions and empirical tests on heterogeneity analysis in 5. Further analysis, according to the results, in high marketization, strong industry competition, as well as non-state-owned enterprises, the positive correlation between network attention and ESG practice is more significant. (Line 360-426)

Comment 6: The contribution of the study to the literature has not been sufficiently mentioned. Regression analysis can go further and more detailed findings can be obtained. Also, further analyses are not mentioned in the method and appear suddenly.

Reply 6: I appreciate it very much for this good suggestion, and I have done it according to your ideas. ① I have reorganized and summarized three possible contributions of this paper. (Line 73-104)

② In the fifth section, labeled as “Further Analysis,” I engage in a comprehensive discussion of the heterogeneity of the results obtained from additional analyses. However, concerning the exposition of the research methodology section, I did not present the specific models(model 1/model 2/model 3) explicitly within the manuscript. Instead, I provided textual elucidations to describe their relevance and application. For instance: I employ the marketization index developed by Wang, Fan and Hu (2019) [60] to determine each location’s marketization level effectively. In Eq. (1), I introduce an interaction term (ATT×Market) to investigate the moderating effect of the marketization process comprehensively. (Line 360-426)

Comment 7: The literature review section is weak and should be developed according to the empirical literature. 

Reply 7: In accordance with the reviewer’s suggestion, I amended the literature review section and supplemented it with crucial literature that I had previously left out.① Literature on the economic consequences of internet concerns. I summarize the research on the economic consequences of online attention, which focuses on the impact on capital markets, and provide examples of the relevant literature. I also added the most relevant literature on the impact of online attention on firms’ behavioural decisions. (Line 106-129)

② In terms of the literature on the influencing factors of ESG, I previously conducted a literature review on the influencing factors of ESG from the perspectives of executive characteristics, business environment, as well as internal and external governance. Now, I have first introduced a wealth of literature on the single dimensional influencing factors of ESG, and pointed out three review articles. Secondly, the comprehensive research literature of ESG was reviewed from the perspectives of macro environment (culture, politics, market environment, etc.), corporate characteristics (ownership, scale, corporate strategy, etc.), corporate governance (equity, board of directors, manager incentives, external governance mechanisms, etc.), as well as many other influencing factors, which greatly supplemented some important literature on the original basis. (Line 130-161)

Comment 8: What is the purpose of choosing this time for the author or authors? Why do not the author or authors go further back?

Reply 8: ① The data on network attention volume comes from the CNRDS (Chinese Research Data Services) database, which provides the network search index values of listed companies in China since 2011 (see Figure 3 of the database introduction, sorry, only the Chinese version is available). The ESG performance data of enterprises is obtained from the Bloomberg database, and the ESG rating of listed companies provided by Bloomberg Consulting has become a relatively well-developed indicator system in China.

② According to the comments of reviewers, I updated the data sample again and selected the data of 2011-2021. There will be slight changes in data in all empirical results (Table 1- Table 6) in the paper, but theoretical derivation will also be supported, and all conclusions in the paper remain unchanged.

Comment 9：What do ST or PT categories mean and what are the long form of ST or PT categories?

Reply 9: ①These PT and ST samples are eliminated to avoid the bias of regression results resulted from special samples. PT is short for Particular Transfer. PT shares refers to shares of listed companies that have suffered losses for three consecutive years, and their shares will be suspended from listing. Such suspended shares will be subject to special transfer services. ST is special treatment, which refers to shares of domestic listed companies that have suffered losses for two consecutive years, and are subject to special treatment. Because there is a mark of whether the company is listed normally in the current year or ST, PT in the China Stock Market & Accounting Research database. (Line 237)

② I will give a footnote on PT and ST enterprises are eliminated from the data cleaning process. (Page 11)

Comment 10: There is no citation although there is a “ Based on the ESG-related literature” phrase. 

Reply 10: I am grateful for the suggestions provided by the reviewer. The selection of control variables in this study is primarily guided by the following prominent references [1,2,3]. In order to adhere to the standards of scholarly writing, I have incorporated two relevant and significant references into the manuscript. (Line 267)

Simultaneously, I will explain why these control variables were chosen. Due to space limitations, I will not include detailed explanations in the main text. The control variables I selected can be divided into the following categories: company characteristics (Size、Ln Age、Lev、Roa), and corporate governance (MS、INST、Top1、Dual、Board、SOE).

Currently, it has been proven by some studies that these control variables an influence ESG . Size [4,5]: Large companies are expected to allocate resources more effectively and show higher ability in ESG. Ln Age [6]. Lev [7]: In companies with good governance, the leverage ratio may be low. ROA [8,9]: Companies with good financial and market performance and profitability have the resources and ability to bear the costs related to ESG investment, and will face higher social constraints and public pressure, which is positively and significantly related to the level of ESG. SOE [10]: State ownership has a positive impact on the level of ESG. Interests of MS management may be inconsistent with those of stakeholders, and the hegemony of management may limit investment in sustainability related issues. INST [11]: The higher the shareholding ratio of INST institutions, the higher the ESG performance. Dual [12]: Power is concentrated in the hands of the CEO, which limits the supervision and control role of other directors and shareholders. The company’s decisions do not always focus on the growth of business value and respect for the wealth of stakeholders. The negative impact of CEO duality on ESG performance. TOP1 [13]: The equity concentration of enterprises is negatively correlated with the degree of ESG. Board [14]: The main empirical results reveal that the board size positively influence a ESG performance while no significant relationship between board average age and ESG performance is found.

[1]Gillan, S. L., A. Koch, and L. T. Starks. 2021. "Firms and Social Responsibility: A Review of Esg and Csr Research in Corporate Finance." Journal of Corporate Finance 66: 101889. doi:10.1016/j.jcorpfin.2021.101889.

[2]Fang, M., Nie, H., & Shen, X. (2023). Can enterprise digitization improve ESG performance?. Economic Modelling, 118, 106101.

[3]Cao, J., W. Li and S. Xiao (2022). "Does mixed ownership reform affect private firms’ ESG practices? Evidence from a quasi‐natural experiment in China." Financial Markets, Institutions & Instruments.

[4]Barros, V., P. V. Matos, J. M. Sarmento, and P. R. Vieira. 2022. "M&a Activity as a Driver for Better Esg Performance." Technological Forecasting and Social Change 175: 121338.

[5]Drempetic, S., Klein, C., & Zwergel, B. (2020). The influence of firm size on the ESG score: Corporate sustainability ratings under review. Journal of Business Ethics, 167(2), 333-360.

[6]Mu, W., K. Liu, Y. Tao, and Y. Ye. 2023. "Digital Finance and Corporate Esg." Finance Research Letters 51: 103426. 

[7]Nadarajah, S., S. Ali, B. Liu, and A. Huang. 2018. "Stock Liquidity, Corporate Governance and Leverage: New Panel Evidence." Pacific-Basin Finance Journal 50: 216-234.

[8]Liu, R., F. Mai, Z. Shan, and Y. Wu. 2020. "Predicting Shareholder Litigation on Insider Trading from Financial Text: An Interpretable Deep Learning Approach." Information & Management 57 (8): 103387. doi:10.1016/j.im.2020.103387.

[9]Sharma, P., P. Panday, and R. Dangwal. 2020. "Determinants of Environmental, Social and Corporate Governance (Esg) Disclosure: A Study of Indian Companies." International Journal of Disclosure and Governance 17: 208-217. 

[10]Al Amosh, H., and S. F. Khatib. 2022. "Ownership Structure and Environmental, Social and Governance Performance Disclosure: The Moderating Role of the Board Independence." Journal of Business and Socio-Economic Development 2 (1): 49-66.

[11]Wang, Y., Y. Lin, X. Fu, and S. Chen. 2023. "Institutional Ownership Heterogeneity and Esg Performance: Evidence from China." Finance Research Letters 51: 103448. 

[12]Romano, M., A. Cirillo, C. Favino, and A. Netti. 2020. "Esg (Environmental, Social and Governance) Performance and Board Gender Diversity: The Moderating Role of Ceo Duality." Sustainability 12 (21): 9298.

[13]Lavin, J. F., and A. A. Montecinos-Pearce. 2021. "Esg Disclosure in an Emerging Market: An Empirical Analysis of the Influence of Board Characteristics and Ownership Structure." Sustainability 13 (19): 10498.

[14]Menicucci, E., & Paolucci, G. (2022). Board Diversity and ESG Performance: Evidence from the Italian Banking Sector. Sustainability, 14(20), 13447. 

Comment 11: Why do the author or authors use regression analysis? Why do the author or authors use regression analysis? How does the literature study this topic? Which method is used for this topic?

Reply 11: My understanding of the reviewer's doubts is as follows: this paper studies the impact of network attention on ESG, multiple regression analysis mainly carried out using unbalanced Panel data of enterprise annual oobservation. This regression analysis method is a commonly adopted research method in the field of corporate finance. For example, regression Analysis is also used in papers published on excellent journals (Journal of Corporate Finance).

Comment 12: It is appropriate to use an alternative indicator for the network attention variable. However, regression analysis may not be sufficient.

Reply 12: In order to enhance the reliability of the research findings, this study also conducted robustness tests. ① In my attempt to address endogeneity concerns using instrumental variable techniques, I have provided additional exposition on the potential endogeneity issues that may arise in this study. I have also provided an explanation for the selection of outstanding shares as instrumental variables and its rationale in this study. ② I have included a supplementary analysis of the robustness test results in this study. (Line 318-355)

Comment 13: These section are highly important for the manuscripts. But there is no enough discussion section and conlusion section is weak too. Mathematical findings are not adequately interpreted. The study, should be written in a more academic language and is open to development. 

Reply 13: I have rewritten the findings and discussion sections. Firstly, the relationship between ESG and CSR has been identified in the discussion of results, indicating that the paper argues that there are differences between the two. Secondly, the drivers of firms' ESG practices are addressed from an informal systems perspective, and the findings in the paper reveal that online attention is a significant external governance mechanism for firms, as is media attention. Thirdly, the possible theoretical contribution of this paper is discussed in terms of the extension of the economic consequences of online attention to the corporate governance literature. Lastly, the conclusions of this paper's study have been further discussed. (Line 428-499)

If the discussion is still inadequate, further revisions will be undertaken. I am looking forward to your valuable comments

Reviewer 2

Comment 1: Information about the method used should be given in the methodology section. Information should be given about how control variables are determined and how they are selected. The control variables used can be supported by attribution.

Reply 1: I’d really appreciate the reviewer’s suggestions. I have endeavoured to amend the various sections of the paper such as the introduction, literature review, theoretical assumptions, data results and discussion of the results in an attempt to make this paper more better.

① In the fifth section, labeled as "Further Analysis," I engage in a comprehensive discussion of the heterogeneity of the results obtained from additional analyses. However, concerning the exposition of the research methodology section, I did not present the specific models(model 1/model 2/model 3) explicitly within the manuscript. Instead, I provided textual elucidations to describe their relevance and application. For instance: “I employ the marketization index developed by Wang, Fan and Hu (2019) [60] to determine each location’s marketization level effectively. In Eq. (1), I introduce an interaction term (ATT×Market) to investigate the moderating effect of the marketization process comprehensively.”

②The selection of control variables in this study is primarily guided by the following prominent references [1,2,3]. In order to adhere to the standards of scholarly writing, I have incorporated two relevant and significant references into the manuscript.

Simultaneously, I will explain why these control variables were chosen. Due to space limitations, I will not include detailed explanations in the main text. The control variables I selected can be divided into the following categories: company characteristics (Size、Ln Age、Lev、Roa), and corporate governance (MS、INST、Top1、Dual、Board、SOE).

Currently, it has been proven by some studies that these control variables an influence ESG . Size [4,5]: Large companies are expected to allocate resources more effectively and show higher ability in ESG. Ln Age [6]. Lev [7]: In companies with good governance, the leverage ratio may be low. ROA [8,9]: Companies with good financial and market performance and profitability have the resources and ability to bear the costs related to ESG investment, and will face higher social constraints and public pressure, which is positively and significantly related to the level of ESG. SOE [10]: State ownership has a positive impact on the level of ESG. Interests of MS management may be inconsistent with those of stakeholders, and the hegemony of management may limit investment in sustainability related issues. INST [11]: The higher the shareholding ratio of INST institutions, the higher the ESG performance. Dual [12]: Power is concentrated in the hands of the CEO, which limits the supervision and control role of other directors and shareholders. The company’s decisions do not always focus on the growth of business value and respect for the wealth of stakeholders. The negative impact of CEO duality on ESG performance. TOP1 [13]: The equity concentration of enterprises is negatively correlated with the degree of ESG. Board [14]: The main empirical results reveal that the board size positively influence a ESG performance while no significant relationship between board average age and ESG performance is found.

[1]Gillan, S. L., A. Koch, and L. T. Starks. 2021. "Firms and Social Responsibility: A Review of Esg and Csr Research in Corporate Finance." Journal of Corporate Finance 66: 101889. doi:10.1016/j.jcorpfin.2021.101889.

[2]Fang, M., Nie, H., & Shen, X. (2023). Can enterprise digitization improve ESG performance?. Economic Modelling, 118, 106101.

[3]Cao, J., W. Li and S. Xiao (2022). "Does mixed ownership reform affect private firms’ ESG practices? Evidence from a quasi‐natural experiment in China." Financial Markets, Institutions & Instruments.

[4]Barros, V., P. V. Matos, J. M. Sarmento, and P. R. Vieira. 2022. "M&a Activity as a Driver for Better Esg Performance." Technological Forecasting and Social Change 175: 121338.

[5]Drempetic, S., Klein, C., & Zwergel, B. (2020). The influence of firm size on the ESG score: Corporate sustainability ratings under review. Journal of Business Ethics, 167(2), 333-360.

[6]Mu, W., K. Liu, Y. Tao, and Y. Ye. 2023. "Digital Finance and Corporate Esg." Finance Research Letters 51: 103426. 

[7]Nadarajah, S., S. Ali, B. Liu, and A. Huang. 2018. "Stock Liquidity, Corporate Governance and Leverage: New Panel Evidence." Pacific-Basin Finance Journal 50: 216-234.

[8]Liu, R., F. Mai, Z. Shan, and Y. Wu. 2020. "Predicting Shareholder Litigation on Insider Trading from Financial Text: An Interpretable Deep Learning Approach." Information & Management 57 (8): 103387. doi:10.1016/j.im.2020.103387.

[9]Sharma, P., P. Panday, and R. Dangwal. 2020. "Determinants of Environmental, Social and Corporate Governance (Esg) Disclosure: A Study of Indian Companies." International Journal of Disclosure and Governance 17: 208-217. 

[10]Al Amosh, H., and S. F. Khatib. 2022. "Ownership Structure and Environmental, Social and Governance Performance Disclosure: The Moderating Role of the Board Independence." Journal of Business and Socio-Economic Development 2 (1): 49-66.

[11]Wang, Y., Y. Lin, X. Fu, and S. Chen. 2023. "Institutional Ownership Heterogeneity and Esg Performance: Evidence from China." Finance Research Letters 51: 103448. 

[12]Romano, M., A. Cirillo, C. Favino, and A. Netti. 2020. "Esg (Environmental, Social and Governance) Performance and Board Gender Diversity: The Moderating Role of Ceo Duality." Sustainability 12 (21): 9298.

[13]Lavin, J. F., and A. A. Montecinos-Pearce. 2021. "Esg Disclosure in an Emerging Market: An Empirical Analysis of the Influence of Board Characteristics and Ownership Structure." Sustainability 13 (19): 10498.

[14]Menicucci, E., & Paolucci, G. (2022). Board Diversity and ESG Performance: Evidence from the Italian Banking Sector. Sustainability, 14(20), 13447. 

I tried our best to improve the manuscript and I appreciate for Editors/Reviewers’ warm work earnestly, and hope that the correction will meet with approval. Once again, thank you very much for your comments and suggestions.

Your sincerely, 

En xie 

bSchool of Management, Hainan University, Haikou 570228, China.Email: xenzi2022@163.com.

---

## [Editor Report · Decision Letter 1]

21 Aug 2023

Can network attention effectively stimulate corporate ESG practices? — Evidence from China

PONE-D-23-10880R1

Dear Dr. En Xie,

We’re pleased to inform you that your manuscript has been judged scientifically suitable for publication and will be formally accepted for publication once it meets all outstanding technical requirements.

Kind regards,

Ercan Özen, PhD

Academic Editor

PLOS ONE
---

## [Editor Report · Acceptance letter]

1 Dec 2023

PONE-D-23-10880R1 

Can network attention effectively stimulate corporate ESG practices? — Evidence from China 

Dear Dr. Xie:

I'm pleased to inform you that your manuscript has been deemed suitable for publication in PLOS ONE. Congratulations! Your manuscript is now with our production department. 

Kind regards, 

on behalf of

Dr. Ercan Özen 

Academic Editor

PLOS ONE